# The Effect of the *BCO2* Genotype on the Expression of Genes Related to Carotenoid, Retinol, and α-Tocopherol Metabolism in Rabbits Fed a Diet with Aztec Marigold Flower Extract

**DOI:** 10.3390/ijms231810552

**Published:** 2022-09-11

**Authors:** Janusz Strychalski, Andrzej Gugołek, Zofia Antoszkiewicz, Dorota Fopp-Bayat, Edyta Kaczorek-Łukowska, Anna Snarska, Grzegorz Zwierzchowski, Angelika Król-Grzymała, Paulius Matusevičius

**Affiliations:** 1Department of Fur-Bearing Animal Breeding and Game Management, University of Warmia and Mazury in Olsztyn, Oczapowskiego 5, 10-719 Olsztyn, Poland; 2Department of Animal Nutrition and Feed Science, University of Warmia and Mazury in Olsztyn, Oczapowskiego 5, 10-719 Olsztyn, Poland; 3Department of Ichthyology and Aquaculture, Faculty of Animal Bioengineering, University of Warmia and Mazury in Olsztyn, Oczapowskiego 5, 10-719 Olsztyn, Poland; 4Department of Microbiology and Clinical Immunology, Faculty of Veterinary Medicine, University of Warmia and Mazury in Olsztyn, Oczapowskiego 13, 10-719 Olsztyn, Poland; 5Department and Clinic of Internal Diseases, Faculty of Veterinary Medicine, University of Warmia and Mazury in Olsztyn, Oczapowskiego 14, 10-719 Olsztyn, Poland; 6Department of Biochemistry, Faculty of Biology and Biotechnology, University of Warmia and Mazury in Olsztyn, Oczapowskiego 1A, 10-719 Olsztyn, Poland; 7Department of Animal Breeding and Nutrition, Faculty of Animal Husbandry Technology, Lithuanian University of Health Sciences, Tilžes 18, LT-47181 Kaunas, Lithuania

**Keywords:** *BCO2* gene, rabbits, marigold, gene expression, carotenoids

## Abstract

This study investigated the effect of the *BCO2* genotype and the addition of Aztec marigold flower extract to rabbit diets on the expression of *BCO1*, *BCO2*, *LRAT*, and *TTPA* genes in the liver. The levels of lutein, zeaxanthin, β-carotene, retinol, and α-tocopherol in the liver and blood serum of rabbits, as well as plasma biochemical parameters and serum antioxidant enzyme activities were also determined. Sixty male Termond White growing rabbits were divided into three groups based on their genotype at codon 248 of the *BCO2* gene (ins/ins, ins/del and del/del); each group was divided into two subgroups: one subgroup received a standard diet, and the other subgroup was fed a diet supplemented with 6 g/kg of marigold flower extract. The obtained results indicate that the *BCO2* genotype may affect the expression levels of *BCO1* and *BCO2* genes in rabbits. Moreover, the addition of marigold extract to the diet of *BCO2* del/del rabbits may increase the expression level of the *BCO2* gene. Finally, an increase in the amount of lutein in the diet of rabbits with the *BCO2* del/del genotype contributes to its increased accumulation in the liver and blood of animals without compromising their health status or liver function.

## 1. Introduction

Carotenoids constitute a lipophilic group of isoprenoid pigments [1]. They are produced by photosynthetic organisms and selected non-photosynthetic species of fungi, bacteria, and algae [2]. More than 750 structured carotenoids have been described to date [3], and they have been divided into two main groups: carotenes (e.g., α-carotene, β-carotene and lycopene) and xanthophylls (e.g., lutein, zeaxanthin and β-cryptoxanthin) [4]. In plants, carotenoids are subsidiary pigments in the photosynthesis process, which protect them against photo-damage and participate in phytohormone biosynthesis [2]. As natural pigments, carotenoids are responsible for the red, orange, pink, and yellow colors of the leaves of plants, fruits, vegetables, and some insects, birds, and fish [5]. Animals and humans are not able to synthesize carotenoids, therefore they must be supplied with food [6]. In animals, carotenoids act as precursors of vitamin A, which is essential for good vision as well as cell growth and differentiation [7]. Due to the highly effective physical and chemical quenching of singlet oxygen and scavenging other reactive oxygen species (ROS), carotenoids are powerful antioxidants in plants and animals. This indicates that carotenoids play a protective role in many ROS-mediated disorders [1]. Carotenoid-rich foods exert beneficial effects in the treatment of diabetes [8], and contribute to regulating the cellular oxidative state in non-alcoholic fatty liver disease (NAFLD) [9], and reducing the risk of lung [10], pharyngeal, laryngeal [11], and colon cancer [12]. Carotenoids are absorbed from the digesta in the intestines and released into the lymph. Next, they are distributed into tissues and organs mostly via (very) low density lipoproteins ((V)LDL) [13]. Dietary carotenoids are accumulated mainly in the liver and adipose tissue [14]. Proper carotenoid metabolism in animals is affected by the interactions between numerous digestive enzymes (e.g., pancreatic lipase and carboxylesterase) and the enzymes involved in carotenoid distribution into various organs and tissues (Pi1 glutathione S-transferase, a protein homologous to the StAR protein) [15,16,17]. Carotenoids are cleaved by two homologous enzymes: BCO1 (β-carotene oxygenase 1), and BCO2 (β-carotene oxygenase 2). Provitamin A carotenoids (with unsubstituted β rings, e.g., α-carotene, β-carotene and β-cryptoxanthin) can be cleaved by both enzymes. It should be noted that BCO2 has a higher affinity for carotenoids that cannot serve as substrates for provitamin A synthesis (without a β ring or with a β ring substituted with a carbonyl, hydroxyl or epoxy group; e.g., lycopene, lutein, and zeaxanthin) [18]. BCO2 catalyzes only the asymmetric oxidative cleavage of carotenoids at the 9,10′ double bond, yielding apo-10′-carotenals and ionones whose functions are still uncertain [19,20]. In turn, BCO1 converts β-carotene to retinal by centric oxidative cleavage at the 15,15′ double bond [21,22,23,24]. Retinal is then reduced to retinol in a reaction catalyzed by retinol dehydrogenases, followed by esterification with fatty acids to form retinyl esters through lecithin:retinol acyl transferase (LRAT) [19]. Most cattle, sheep, and rabbits have fat that can be visually assessed as white at slaughter. However, some representatives of these species are characterized by a dysfunction in the BCO2 protein, due to which non-provitamin A carotenoids are not cleaved and are stored in the adipose tissue, giving it a yellow color, although such cases are rare [25,26,27]. Yellow fat is also often seen in poultry [28]. In rabbits, the deletion of AAT nucleotides at codon 248 of the *BCO2* gene gives a yellow tint to fat [26]. The yellow color of fat is a recessive trait observed only in homozygous individuals that carry the above deletion [26,29,30].

Marigold flowers are a valuable source of carotenoids. The Aztec marigold (*Tagetes erecta* L.) belongs to the family *Asteraceae* and is native to Mexico. It is an annual plant grown in Europe, Asia, and Africa for ornamental, medicinal, and industrial purposes [31]. Lutein, the main pigment present in the petals of marigold flowers, accounts for nearly 90% of total carotenoids. Zeaxanthin constitutes approximately 5% of the carotenoids found in marigold flowers [32,33]. Aztec marigold flower extract in the form of dried powder is widely used as an additive to poultry feed to improve the pigmentation of the birds’ fat, skin and egg yolk [31].

Our previous studies analyzed the levels of carotenoids, retinol, and α-tocopherol in the adipose tissue and liver of rabbits and in the milk of rabbit depending on their genotype determined by AAT-deletion mutation at codon 248 of the *BCO2* gene. Considering that a diet rich in carotenoids increases yellow pigmentation in animals [34] and that there is a strong correlation between carotenoid consumption and plasma concentration [35], this study was undertaken to investigate the effect of the *BCO2* genotype and the addition of Aztec marigold flower extract to rabbit diets on the expression of genes related to the metabolism of carotenoids and vitamins A and E (*BCO1*, *BCO2*, *LRAT*, and *TTPA*) in the liver. The levels of lutein, zeaxanthin, β-carotene, retinol, and α-tocopherol in the liver and blood serum of rabbits were also determined, as well as plasma biochemical parameters and serum antioxidant enzyme activities.

## 2. Results

Genotype at codon 248 of the *BCO2* gene was associated with changes in the expression of the *BCO1* gene in rabbits fed the control diet (Figure 1). The relative expression level of the *BCO1* gene was lowest in homozygous ins/ins rabbits, and highest in heterozygous ins/del animals (*p* < 0.01). An intermediate level of *BCO1* gene expression was noted in rabbits with the deletion of both *BCO2* alleles. Genotype at codon 248 of the *BCO2* gene did not induce changes in the expression of the remaining genes in rabbits fed the control diet.

Genotype at codon 248 of the *BCO2* gene was associated with changes in the expression of the *BCO2* gene in rabbits fed a diet supplemented with 6 g/kg of Aztec marigold flower extract (Figure 2). The relative expression level of the *BCO2* gene was lowest in homozygous ins/ins rabbits, and highest in homozygous del/del animals (*p* < 0.05). An intermediate level of *BCO2* gene expression was noted in heterozygous rabbits. Genotype at codon 248 of the *BCO2* gene did not induce changes in the expression of the remaining genes in rabbits fed the marigold-supplemented diet.

Dietary supplementation with 6 g/kg of Aztec marigold flower extract did not affect the expression of the analyzed genes in rabbits with the ins/ins genotype or the ins/del genotype at codon 248 of the *BCO2* gene (Appendix A). In turn, the expression of the *BCO2* gene increased in rabbits with the deletion of both *BCO2* alleles (*p* < 0.01, Figure 3), and the expression of the remaining genes remained unchanged.

Liver lutein concentrations were many times higher in rabbits with double deletion at codon 248 of the *BCO2* gene (del/del) than in those with ins/ins and ins/del genotypes (Table 1). The above applied to animals fed both the control diet and the marigold-supplemented diet (*p* < 0.01 in both cases). Moreover, the addition of marigold extract to the diet of rabbits with the del/del genotype caused a four-fold increase in their liver lutein levels (*p* < 0.01), compared with their counterparts fed the control diet. Differences were also found in β-carotene concentrations in the liver of rabbits. In the control group, the levels of this micronutrient were higher in homozygous del/del rabbits than in those with ins/ins and ins/del genotypes (*p* < 0.05); in the marigold group, a significant difference was noted between rabbits with the del/del genotype and those with the ins/del genotype (0.61 μg/g vs. 0.32 μg/g, respectively, *p* < 0.05).

The *BCO2* genotype affected the serum levels of lutein and β-carotene (Table 2). Lutein content was higher in homozygous del/del rabbits than in animals with ins/ins and ins/del genotypes, but only in the marigold group (del/del 2.01 μg/g vs. ins/ins 0.68 μg/g and ins/del 0.69 μg/g, *p* < 0.01 in both cases). The same relationships were observed for β-carotene levels in the groups fed both diets (*p* < 0.01 in both cases). Moreover, the diet affected the serum concentrations of lutein, zeaxanthin, and β-carotene. Rabbits with the del/del genotype fed the marigold-supplemented diet were characterized by higher levels of lutein and zeaxanthin than their counterparts fed the control diet (*p* < 0.05 in both cases). A similar relationship was noted for β-carotene content in homozygous rabbits (*p* < 0.05).

*BCO2* gene polymorphism affected the plasma levels of ALP and LDH (Table 3). In the marigold group, plasma ALP levels were lower in rabbits with the ins/del genotype than in those with the del/del genotype (*p* < 0.01). Plasma LDH levels were lower in homozygous del/del rabbits than in animals with the ins/ins genotype (*p* < 0.01 in both groups). Rabbits with the del/del genotype fed the marigold-supplemented diet had lower LDH levels than their counterparts fed the control diet (79.00 U/L vs. 123.20 U/L, *p* < 0.05).

No significant differences were found in the activity of antioxidant enzymes in the blood serum of rabbits (Table 4). In the marigold group, SOD activity reached 393.5 IU/mL, 411.0 IU/mL, and 252.5 IU/mL in rabbits with ins/ins, ins/del, and del/del genotypes, respectively (*p* = 0.070).

## 3. Discussion

The present study investigated the expression of genes related to the metabolism of carotenoids and vitamin A with regard to the excess of carotenoids in feedstuffs offered to male Termond White rabbits. It was hypothesized that the surplus amounts of lutein, zeaxanthin, β-carotene, retinol, and α-tocopherol will not affect the health status of the host or liver function. Previous research [29] has shown that polymorphism of the *BCO2* gene differentiates lutein content in the liver and carotenoid concentrations in perirenal fat in various crossbred rabbits. However, the question whether the dietary inclusion rates of carotenoids affect their accumulation in the body remained unanswered. To the best of our knowledge, this is the first study involving rabbits with various genotypes at codon 248 of the *BCO2* gene fed diets with different inclusion levels of carotenoids. Specifically, the control group received 21.64 mg/kg of lutein, whereas the marigold group received almost 320 mg/kg of lutein. Such an approach allowed us to determine changes in the expression of genes related to the metabolism of carotenoids and vitamins A and E in the liver (1), the rates of carotenoid, retinol, and α-tocopherol accumulation in the liver and blood serum (2), and blood plasma biochemical parameters and the activity of antioxidant enzymes (3).

In the control group, rabbits with the ins/del genotype in the *BCO2* gene were characterized by higher expression of the *BCO1* gene than the animals with the ins/ins genotype. A similar tendency in *BCO1* expression, although not statistically significant, was observed in marigold group rabbits. This phenomenon can be attributed to the role played by the *BCO1* gene. Specifically, BCO1 is a β-carotene cleavage enzyme leading to the formation of vitamin A. Thus, it is not surprising that in this study, the increased level of *BCO1* expression in animals with the ins/del genotype in the *BCO2* gene fed the control diet corresponded to the relatively lowest level of β-carotene and the relatively highest level of retinol in the liver. According to Lietz et al. [20], *BCO1* activity can be affected by fats, proteins, antioxidants, carotenoids, and vitamin A status. This activity seems to be species-dependent. Xanthophyll supplementation at 20 mg/kg and 40 mg/kg had no effect on *BCO1* mRNA expression in the liver of hens and chicks [36]. It should be noted, however, that poultry feed usually contains lower amounts of carotenoids than rabbit feed, and therefore the experimental addition of xanthophyll was much lower in poultry than in rabbits.

In the marigold group, rabbits with the del/del genotype in the *BCO2* gene were characterized by higher expression of the *BCO2* gene than the animals with the ins/ins genotype. The observed phenomenon is probably dose-dependent because such a relationship was not noted in control group rabbits. The obtained results are complemented by the fact that rabbits with the deletion of both alleles, fed the marigold-supplemented diet, exhibited higher *BCO2* gene expression than those receiving the control diet. As *BCO2* del/del rabbits are able to absorb but not degrade xanthophylls, it is obvious that rabbits with this genotype belonging to the marigold group had the highest liver and serum lutein levels.

In the current study, neither the *BCO2* genotype nor the diet affected the expression of genes associated with the metabolism of vitamins A and E (*LRAT* and *TTPA*, respectively), which is consistent with the absence of intergroup differences in the amounts of these vitamins in the liver. As reported by Gao et al. [36], xanthophyll supplementation reduced *LRAT* expression in the liver of hens, but at the same time it increased the expression of another gene playing an important role in retinoid metabolism - *STRA6* (signaling receptor and transporter of retinol). This indicates the need for more comprehensive gene expression studies, including several genes involved in selected metabolic pathways.

In the current study, lutein concentrations in the liver and serum were higher in del/del rabbits, compared with the other genotypes, regardless of the diet. In our previous research [29], hepatic lutein concentrations in the control group were 0.57, 0.67, and 2.76 in *BCO2* ins/ins, ins/del, and del/del rabbits, respectively, and they were higher than those obtained in the present study. However, the cited study involved rabbits raised until 140 days of age and fed a diet containing 37.70 mg/kg of lutein, whereas in the current experiment, rabbits were raised until 91 days of age and their basal diet had lower lutein content (21.64 mg/kg). In the control group, serum lutein levels were 1.80-fold and 1.48-fold higher in rabbits with the del/del genotype than in those with ins/ins and ins/del genotypes, respectively. Even more pronounced differences were noted in the marigold group where serum lutein levels were 2.96-fold and 2.91-fold lower in rabbits with ins/ins and ins/del genotypes than in del/del genotype animals, respectively. From the biochemical point of view, lutein exhibits substantial antioxidant activity in in vitro studies [37]. Lutein is beneficial for vision and eye health [38], and it exerts anticancer and anti-inflammatory effects [39]. Kiplimo et al. [40] reported that lutein suppressed the growth and proliferation of multiple pathogenic bacterial strains, including *E. coli*, *S. aureus*, and *P. aeruginosa*, thus improving the health status of the host. Chew et al. [41] also described lutein as a lymphocyte proliferation-promoting agent. Park et al. [42] demonstrated that supplemental dietary lutein from marigold flowers was detected in serum soon after ingestion, and it was subsequently stored in the liver. The current findings corroborate the results of our previous studies [29,30,43], which revealed that the del/del genotype is associated with increased hepatic lutein concentrations irrespective of dietary lutein content.

Interestingly, no intergroup differences were found in zeaxanthin concentrations in the liver of rabbits, which could result from the poor assimilation of xanthophylls [44]. The assimilation of biochemically active compounds is a multistage and complex process; first, zeaxanthin is released from the food matrix and transported, in the form of emulsion, to the duodenum where it is released by pancreatic enzymes, absorbed by enterocytes and utilized in the Golgi apparatus in the process of HDL fraction assembly [45]. The latter is released into the bloodstream, leading to zeaxanthin accumulation in the liver, adipose tissue, and other tissues, mainly retina [46,47].

Rabbits with the del/del genotype in the *BCO2* gene, fed the marigold-supplemented diet, were characterized by the lowest level of LDH and the highest level of ALP in the blood plasma. In the literature, a decrease in LDH concentration is not considered an important finding [48], but an increase in ALP levels may be associated with liver disease indicative of bile stasis related to e.g., lipidosis [43]. However, elevated ALP levels are often reported in clinically healthy rabbits, which is due to the fact that they produce two ALP isoenzymes and one intestinal isoenzyme in the liver; therefore, serum ALP concentrations are actually the sum of these three isoenzymes [49]. Moreover, rabbits are often raised on small backyard farms where they receive large amounts of green fodder containing several hundred mg/kg of xanthophylls [50], and there are no reports of disease manifestations in rabbits with the deletion of both *BCO2* alleles, including wild rabbits [29,30,51,52]. This suggests that elevated ALP levels in rabbits probably are not a cause for concern.

A novel finding of the present study is that an increase in the amount of lutein to 320 mg/kg in the diet of rabbits with the *BCO2* del/del genotype contributes to increased accumulation of this micronutrient in the liver and blood of animals without compromising their health status or liver function. This information can be valuable for both rabbit breeders and rabbit meat consumers. Another implication of the study is that the *BCO2* genotype may affect the expression levels of *BCO1* and *BCO2* genes in rabbits. Moreover, the addition of supplemental lutein to the basal diet of *BCO2* del/del rabbits may increase the expression level of the *BCO2* gene. These results may contribute to a better understanding of carotenoid metabolism in rabbits and, possibly, in other animal species. The present study has certain limitations, which may, however, pave the way for future research. Since this is the first study to determine changes in the expression of selected genes, depending on the *BCO2* genotype of rabbits and the dietary inclusion levels of carotenoids, the obtained results should be confirmed in experiments involving the use of different diets, different rabbit breeds and, possibly, specialized cell lines. Moreover, gene expression was analyzed based on liver samples only, and only one gene from each metabolic pathway was examined. Therefore, the obtained data are fragmentary. Future research should focus on other tissues and organs, such as adipose tissue, eyes and the brain, and it should include more genes involved in the studied metabolic pathways.

## 4. Materials and Methods

### 4.1. Animals and Diets

All experimental procedures were approved by the Local Ethics Committee in Olsztyn, Poland (consent No. 065.12.2020), and the animals were cared for in accordance with the guidelines of the European Parliament (EC Directive 86/60963/2010 on the protection of animals used for experimental and other scientific purposes).

Sixty male Termond White growing rabbits raised in a closed facility on a farm in northern Poland were used in this study. Rabbits were divided into three experimental groups (*n* = 20, body weight of 726.1 g ± 66.09, mean ± SD) based on their genotype at codon 248 of the *BCO2* gene (ins/ins, ins/del and del/del). In this article, the gene variant with the AAT-deletion at codon 248 of the *BCO2* gene is referred to as “del,” and the variant without the above deletion is referred to as “ins”. In order to determine the effect of the diet on carotenoid intake and/or storage, each experimental group was divided into two subgroups; one subgroup received a standard diet (control group, *n* = 10), and the other subgroup was fed a diet supplemented with 6 g/kg of Aztec marigold flower extract in the form of dried powder containing 4.96% of lutein and 0.28% of zeaxanthin (marigold group, *n* = 10). The ingredients and chemical composition of the control diet are presented in Table 5.

The experiment lasted for 8 weeks, from weaning at 35 days of age (June) until 91 days of age (August). Rabbits were housed individually under standard conditions, in wire-mesh flat-deck cages, which supported their visual contact with other rabbits. Temperature and relative air humidity were maintained within the range of 16–18 °C and 60–75%, respectively. The photoperiod was fully controlled, with 12 h of light and 12 h of darkness. The experimental facility had a forced ventilation system. At the end of the experiment, the animals were slaughtered by a veterinarian. They were bled within two minutes after physical stunning, and blood samples were collected into 2.5 mL heparinized test tubes. Immediately after skinning and evisceration, approximately 20 g samples of liver tissues were collected for chemical analyses. In order to determine RNA expression levels, liver samples were also snap frozen in liquid nitrogen immediately after slaughter, and were stored at −80 °C until RNA extraction.

### 4.2. Genotyping

Genomic DNA was isolated from hair roots. Insertion/deletion polymorphism at codon 248 in the *BCO2* gene was determined by PCR-RFLP, as described by Strychalski et al. [26]. Briefly, primers encompassing *BCO2* intron 5 and exon 6, forward (5′GGGAATTCATCTGCATGGAC3′) and reverse (5′TCGATTCTGCAGGAGCAATA3′), were used for PCR. A total of 10 μL of the PCR product was digested with 10 U of AanI (Thermo Fisher Scientific, Waltham, MA, USA), and restriction fragments were separated by electrophoresis on 2.2% agarose gel with ethidium bromide.

### 4.3. RNA Expression

Approximately 100 µg of frozen liver samples were used for RNA isolation with the use of TRIzolTM Reagent (Thermo Fisher Scientific, Waltham, MA, USA), according to the manufacturer’s protocol. The concentrations of eluted RNA were measured with the NanoDrop 2000 spectrophotometer (Thermo Fisher Scientific, Waltham, MA, USA). The samples were stored at −80 °C for further analysis. Genomic DNA that remained in the samples after RNA isolation was digested with DNase I (Invitrogen, Waltham, MA, USA). Reverse transcription was carried out with the RevertAid First Strand cDNA Synthesis Kit (Thermo Fisher Scientific, Waltham, MA, USA), in accordance with the manufacturer’s instructions. The concentration of RNA for the synthesis of complementary DNA (cDNA) was standardized to 0.5 μg per sample. The expression of the gene encoding *BCO1*, *BCO2*, *LRAT*, and *TTPA* was determined by real-time PCR (LightCycler^®^ 96, Roche, Basel, Switzerland) with DyNAmo HS SYBR Green (Life Technologies, Carlsbad, CA, USA). *GAPDH* and *β-actin* were used as reference genes. About 2 μL of cDNA was used for each PCR reaction. The list of primers with sequences, annealing temperature, and amplification efficiency is presented in Table 6. The reaction was carried out under the following conditions: polymerase activation at 95 °C for 15 min, followed by 40 two-stage cycles: denaturation at 94 °C for 10 s, primer annealing at 55 °C for 20 s, and chain elongation at 72 °C for 30 s.

### 4.4. Blood Plasma Biochemical Parameters and Serum Antioxidant Enzyme Activities

Blood plasma was obtained by centrifugation (380× *g*, MPW-352R, MPW MED. Instruments, Warsaw, Poland) of heparinized blood samples, and it was stored at −70 °C until analyses. The following plasma biochemical parameters were determined on an automatic analyzer (Pentra C200, Horiba Ltd., Kyoto, Japan): the concentrations of total protein and albumin, and the activity of gamma-glutamyl transferase (GGT), asparagine aminotransferase (AST), alanine aminotransferase (ALT), lactate dehydrogenase (LDH) and alkaline phosphatase (ALP). The following antioxidant enzymes were determined (ACCENT-200 biochemistry analyzer, Cormay, Warsaw, Poland): glutathione peroxidase (POX) and superoxide dismutase (SOD).

### 4.5. Chemical Analyses

#### 4.5.1. Proximate Chemical Composition and Energy Content of Feed

All analyses were carried out in duplicate. The nutrient content of feed was determined by standard methods [53]. Dry matter content was determined in a laboratory drier, at 103 °C. Crude ash content was estimated by sample mineralization in a muffle furnace (Czylok, Jastrzębie-Zdrój, Poland) at 600 °C. Total nitrogen content was determined by the Kjeldahl method, in the Foss Kjeltec 2200 Auto Distillation Unit (Foss, Warsaw, Poland). Ether extract content was estimated by the Soxhlet method, in the Foss Soxtec System 2043 (Foss, Warsaw, Poland). Neutral detergent fiber (NDF) and acid detergent fiber (ADF) were analyzed in the Foss Tecator Fibertec 2010 System (Foss, Warsaw, Poland); NDF was determined as described by Van Soest et al. [54], and ADF was determined according to the official methods of analysis of AOAC International [53]. Gross energy content was determined in a bomb calorimeter (C2000 basic, IKA, Staufen, Germany).

#### 4.5.2. Content of Retinol, α-Tocopherol and Carotenoids in Feed and in the Liver and Blood Serum of Rabbits

The concentrations of retinol and α-tocopherol in feed were determined by high-performance liquid chromatography (HPLC) in accordance with Polish Standards (PN-EN ISO 14565 2002; PN–EN ISO 6867 2002). Analytical samples were saponified (50% KOH in ethanol *w*/*v*, 90 °C, 30′, reflux condenser) and extracted with ethanol and petroleum ether. The extracts (upper fraction–hydrophobic ether extract; lower fraction–hydrophilic) were rinsed with a 10% (*w*/*v*) aqueous solution of NaCl followed by distilled water, dehydrated with anhydrous sodium sulfate, and evaporated to dryness under a stream of nitrogen (N2). The residue was dissolved in 96% ethanol to determine the content of tocopherols and retinol, and in n-hexane to determine the content of β-carotene and xanthophylls.

The concentrations of retinol and α-tocopherol in animal tissues were determined as described by Hőgberg et al. [55] and Xu [56]. One gram samples of liver and adipose tissue were homogenized with 1.2 cm^3^ of 20% ascorbic acid in H2O (Ultra-Turrax T25 homogenizer, IKA, Staufen, Germany) and saponified (375 g KOH, 750 cm^3^ H2O, 450 cm^3^ methanol, 80 °C, 15′). Ethanol (35% *v*/*v*) and NaCl (10% *w*/*v*) were added to cooled samples, which were extracted twice with n-hexane (2 × 4 cm^3^) and centrifuged (MPW-350R centrifuge, Poland, 1500 g, 15′). Next, 3 cm^3^ and 4 cm^3^ samples of the supernatant were collected successively. The combined extracts were evaporated to dryness under a stream of nitrogen. The residue was dissolved in 1 cm^3^ of 96% ethanol to determine the content of tocopherols and retinol, and in n-hexane to determine the content of β-carotene and xanthophylls.

The levels of retinol and α-tocopherol in feed, liver and blood serum were determined by reversed-phase (RP) HPLC (HPLC system, SHIMADZU, Kyoto, Japan), on a Nucleosil C18 column, with methanol/water (95:5, *v*:*v*) as the mobile phase, a UV detector (326 nm for retinol) and a fluorescence detector (excitation–295 nm, emission–330 nm for tocopherol), flow rate: 1 cm^3^ min^−1^, loop: 20 μl. The concentrations of retinol and tocopherol were estimated by comparing their peak areas with those of external standards (Sigma-Aldrich, Darmstadt, Germany: (±)-α-tocopherol–No Cat. T3251, retinol-vitamin A–alcohol-No Cat. R7632; the recoveries of α-tocopherol and retinol were 96% ± 2.8% and 89% ± 4.5%, respectively).

The hexane or petroleum ether extract of carotenoids (β-carotene and xanthophylls) was filtered through a 22 μm PTFE syringe filter (30-SF-02 CHROMACOL Ltd.) and analyzed by RP-HPLC (SHIMADZU, Kyoto, Japan), column: Gemini 5μm (Phenomenex), C18, 110 Å, 250 x4 mm; mobile phase: methanol:tetrahydrofuran (95:5 *v*/*v*) (HPLC grade Sigma-Aldrich, Darmstadt, Germany); flow rate: 1 cm^3^ min^−1^, detector: UV-vis 450 nm; loop: 20 µL. External standards: β-carotene (β-carotene, provitamin A-No Cat. C4646; β-carotene recovery–85% ± 1.9%) and xanthophyll (xanthophyll–lutein, α-carotene-3,3′-diol; No Cat. X6250; xanthophyll recovery–86% ± 6%) standards from Sigma-Aldrich (Darmstadt, Germany) [57].

### 4.6. Statistical Analysis

All calculations were performed using R software [58]. In order to avoid useless comparisons, a one-way design model was used where three different genotypes were compared within the same diet and two different diets were compared within the same genotype.

The results of qPCR were analyzed using the specialized MCMC.qpcr package [59] where a Poisson-lognormal generalized mixed model is implemented to a complete set of qPCR measurements (adjusted for amplification efficiency) using a Markov Chain Monte Carlo (MCMC) procedure. This procedure infers changes in the expression of all genes from the joint posterior distribution of parameters. The reference genes (*GAPDH* and *β-actin*) were added as priors to the model function to account for variations in cDNA, and were allowed 1.2-fold changes [59]. In charts, posterior means and 95% credible intervals were plotted as log2 abundance. The 95% credible intervals are analogous to Bayesian 95% confidence intervals and are included to indicate the extent of variability in the posterior distribution.

The remaining data, i.e., the content of selected compounds in the liver and blood serum of rabbits, the values of blood plasma biochemical parameters, and serum antioxidant enzyme activities, were processed statistically by analysis of variance (ANOVA) followed by Duncan’s multiple range test. These data are presented in the tables as means ± standard deviations (SD).

## Figures and Tables

**Figure 1 ijms-23-10552-f001:**
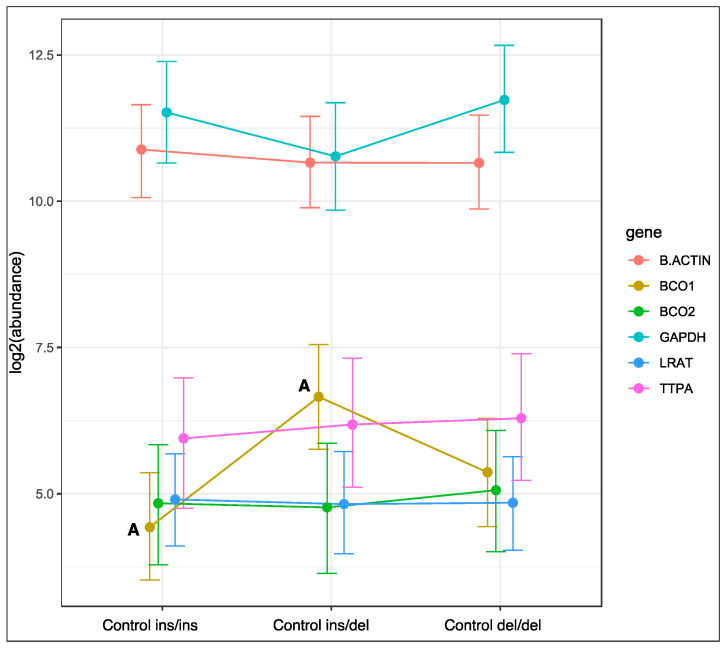
Relative *BCO1*, *BCO2*, *LRAT*, and *TTPA* mRNA levels in the liver of rabbits with different genotypes at codon 248 of the *BCO2* gene (ins/ins vs. ins/del vs. del/del) fed the control diet. *GAPDH* and *β-actin* were used as reference genes. Data represent posterior means (expressed as arbitrary units) ± 95% credible intervals. Means followed by the same superscripts (A, A) differ highly significantly (*p* ≤ 0.01).

**Figure 2 ijms-23-10552-f002:**
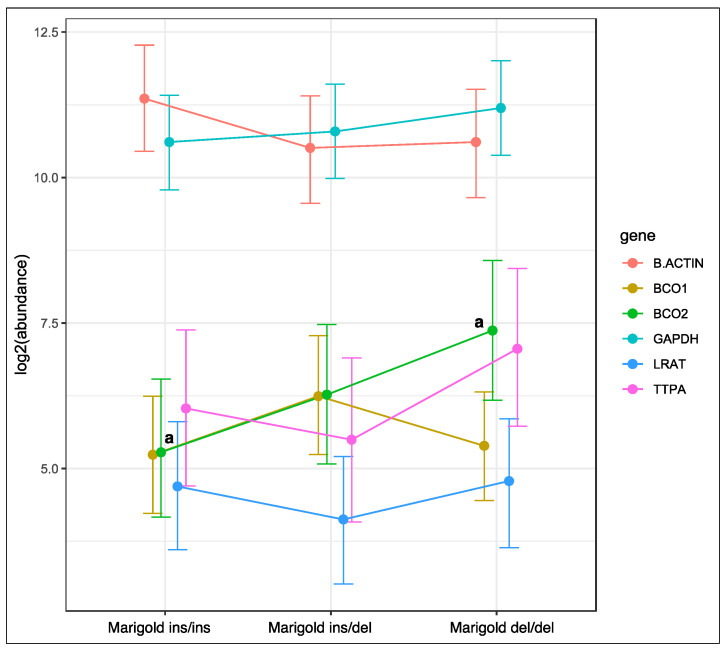
Relative *BCO1*, *BCO2*, *LRAT*, and *TTPA* mRNA levels in the liver of rabbits with different genotypes at codon 248 of the *BCO2* gene (ins/ins vs. ins/del vs. del/del) fed a diet with the addition of Aztec marigold flower extract. *GAPDH* and *β-actin* were used as reference genes. Data represent posterior means (expressed as arbitrary units) ± 95% credible intervals. Means followed by the same superscripts (a, a) differ significantly (*p* ≤ 0.05).

**Figure 3 ijms-23-10552-f003:**
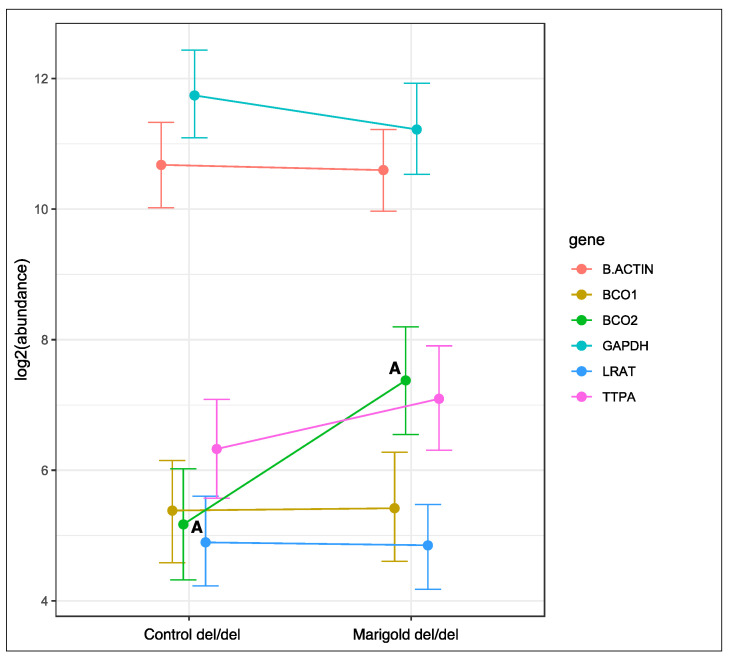
Relative *BCO1*, *BCO2*, *LRAT*, and *TTPA* mRNA levels in the liver of rabbits with the del/del genotype at codon 248 of the *BCO2* gene, fed different diets (control diet vs. a diet with the addition of Aztec marigold flower extract). *GAPDH* and *β-actin* were used as reference genes. Data represent posterior means (expressed as arbitrary units) ± 95% credible intervals. Means followed by the same superscripts (A, A) differ highly significantly (*p* ≤ 0.01).

**Table 1 ijms-23-10552-t001:** Content of selected compounds in the liver (μg/g) of rabbits (mean ± SD).

Compound	Diet	*BCO2* Genotypes			*p*-Value
		ins/ins	ins/del	del/del	
Lutein	Control	0.31 ± 0.16 ^A^	0.40 ± 0.28 ^B^	1.94 ± 1.49 ^AB^	<0.001
	Marigold	0.42 ± 0.32 ^A^	0.50 ± 0.32 ^B^	8.01 ± 6.64 ^AB^	<0.001
*p*-value		0.428	0.452	0.006	
Zeaxanthin	Control	0.13 ± 0.04	0.13 ± 0.10	0.17 ± 0.09	0.393
	Marigold	0.15 ± 0.11	0.15 ± 0.12	0.20 ± 0.12	0.587
*p*-value		0.279	0.344	0.792	
β-carotene	Control	0.25 ± 0.13 ^a^	0.22 ± 0.11 ^b^	0.46 ± 0.25 ^ab^	0.016
	Marigold	0.35 ± 0.20	0.32 ± 0.23 ^a^	0.61 ± 0.43 ^a^	0.042
*p*-value		0.422	0.376	0.318	
Retinol	Control	95.99 ± 36.00	102.22 ± 49.75	93.01 ± 63.65	0.401
	Marigold	101.93 ± 86.40	98.56 ± 57.42	97.53 ± 51.42	0.988
*p*-value		0.843	0.917	0.882	
α-tocopherol	Control	6.13 ± 4.83	6.44 ± 5.77	6.97 ± 3.35	0.924
	Marigold	5.86 ± 4.50	5.67 ± 4.35	8.60 ± 7.33	0.274
*p*-value		0.879	0.729	0.529	

Means within a row followed by the same superscripts (A, A; B, B) differ highly significantly (*p* ≤ 0.01). Means within a row followed by the same superscripts (a, a; b, b) differ significantly (*p* ≤ 0.05).

**Table 2 ijms-23-10552-t002:** Content of selected compounds in the blood serum (μg/g) of rabbits (mean ± SD).

Compound	Diet	*BCO2* Genotypes			*p*-Value
		ins/ins	ins/del	del/del	
Lutein	Control	0.51 ± 0.31	0.62 ± 0.47	0.92 ± 0.95	0.406
	Marigold	0.68 ± 0.36 ^A^	0.69 ± 0.57 ^B^	2.01 ± 1.36 ^AB^	<0.001
*p*-value		0.267	0.762	0.013	
Zeaxanthin	Control	0.03 ± 0.02	0.03 ± 0.02	0.03 ± 0.02	0.996
	Marigold	0.04 ± 0.03	0.05 ± 0.04	0.07 ± 0.05	0.255
*p*-value		0.250	0.065	0.016	
β-carotene	Control	0.03 ± 0.02 ^A^	0.02 ± 0.01 ^B^	1.40 ± 0.65 ^AB^	<0.001
	Marigold	0.04 ± 0.02 ^A^	0.05 ± 0.02 ^B^	1.73 ± 1.50 ^AB^	<0.001
*p*-value		0.846	0.020	0.523	
Retinol	Control	1.41 ± 1.30	2.61 ± 1.67	2.78 ± 1.03	0.066
	Marigold	1.91 ± 1.50	3.88 ± 2.50	2.29 ± 1.18	0.052
*p*-value		0.434	0.198	0.340	
α-tocopherol	Control	2.37 ± 1.35	2.36 ± 1.97	2.29 ± 1.90	0.993
	Marigold	3.03 ± 1.37	2.66 ± 1.89	2.17 ± 1.34	0.389
*p*-value		0.288	0.736	0.854	

Means within a row followed by the same superscripts (A, A; B, B) differ highly significantly (*p* ≤ 0.01).

**Table 3 ijms-23-10552-t003:** Blood plasma biochemical parameters of rabbits (mean ± SD).

Compound	Diet	*BCO2* genotypes			*p*-Value
		ins/ins	ins/del	del/del	
AST (U/L)	Control	14.70 ± 5.68	15.20 ± 6.94	20.10 ± 5.36	0.106
	Marigold	20.30 ± 9.25	14.10 ± 5.45	16.30 ± 8.46	0.223
*p*-value		0.120	0.698	0.246	
ALT (U/L)	Control	8.50 ± 1.78	7.30 ± 4.57	6.10 ± 1.79	0.224
	Marigold	7.60 ± 4.48	6.20 ± 3.82	6.60 ± 2.17	0.677
*p*-value		0.562	0.567	0.581	
GGT (U/L)	Control	23.80 ± 10.01	24.10 ± 9.48	23.50 ± 12.42	0.992
	Marigold	22.20 ± 15.08	20.90 ± 8.72	19.40 ± 6.77	0.855
*p*-value		0.770	0.442	0.371	
Albumin (g/L)	Control	33.50 ± 14.50	36.94 ± 14.27	36.92 ± 16.58	0.844
	Marigold	38.98 ± 10.77	37.25 ± 13.02	37.32 ± 10.67	0.931
*p*-value		0.350	0.960	0.950	
ALP (U/L)	Control	6.98 ± 2.77	8.00 ± 5.64	9.17 ± 5.15	0.586
	Marigold	10.20 ± 5.48	5.90 ± 1.98 ^A^	14.82 ± 8.48 ^A^	0.009
*p*-value		0.115	0.281	0.088	
LDH (U/L)	Control	220.20 ± 55.31 ^A^	179.70 ± 77.20	123.20 ± 40.39 ^A^	0.004
	Marigold	213.70 ± 93.22 ^A^	136.30 ± 72.00	79.00 ± 45.51 ^A^	0.001
*p*-value		0.852	0.210	0.034	
Total protein (g/L)	Control	54.60 ± 23.65	59.76 ± 29.72	55.78 ± 21.58	0.892
	Marigold	57.05 ± 17.42	57.71 ± 35.66	54.19 ± 23.88	0.952
*p*-value		0.795	0.890	0.878	

Means within a row followed by the same superscripts (A, A) differ highly significantly (*p* ≤ 0.01).

**Table 4 ijms-23-10552-t004:** Activity of antioxidant enzymes in the blood serum (IU/mL) of rabbits.

Compound	Diet	*BCO2* Genotypes			*p*-Value
		ins/ins	ins/del	del/del	
POX	Control	1065 ± 438	1037 ± 451	1237 ± 612	0.639
	Marigold	996 ± 361	1235 ± 580	1197 ± 527	0.668
*p*-value		0.847	0.528	0.929	
SOD	Control	415.9 ± 149.7	342.2 ± 186.5	355.1 ± 147.9	0.563
	Marigold	393.5 ± 139.5	411.0 ± 190.9	252.5 ± 145.8	0.070
*p*-value		0.733	0.426	0.136	

**Table 5 ijms-23-10552-t005:** Ingredients, chemical composition and energy content of the control diet.

Ingredients (%)		Chemical Composition and Energy Content	
Dried alfalfa	33.5	Dry matter (%)	89.15
Oat	17	Crude protein (%)	16.68
Wheat bran	15	Ether extract (%)	3.20
Barley	12.3	Crude ash (%)	6.74
Wheat	9	Neutral detergent fiber (%)	26.63
Soyabean meal	8	Acid detergent fiber (%)	18.57
Linseed	2	Gross energy (MJ/kg)	17.65
Mineral supplements ^a^	1.8	Lutein (mg/kg)	21.64
Mineral–vitamin premix ^b^	1	Zeaxanthin (mg/kg)	0.14
NaCl	0.4	β-carotene (mg/kg)	9.97
Total	100	Retinol (IU/kg)	9734
		α-tocopherol (mg/kg)	20.71

^a^ Mineral supplements: calcium carbonate, dicalcium phosphate, sodium hydrogen carbonate. ^b^ Mineral–vitamin premix (1 kg): vitamin A: 3,500,000 IU, vitamin D_3_: 200,000 IU, vitamin E: 28,000 mg, vitamin K_3_: 200 mg, vitamin B_1_: 1500 mg, vitamin B_2_: 2800 mg, vitamin B_6_: 2800 mg, vitamin B_12_: 20,000 mcg, folic acid: 200 mg, niacin: 10,000 mg, biotin: 200,000 mcg, calcium pantothenate: 7000 mg, choline: 30,000 mg, Fe: 17,000 mg, Zn: 2000 mg, Mn: 1000 mg, Cu (copper sulphate × 5H_2_O. 24.5%): 800 mg, Co: 1000 mg, I: 100 mg, methionine: 150 g, Ca: 150 g, P: 100 g.

**Table 6 ijms-23-10552-t006:** RT-qPCR primers.

Gene	Sequence (5′–3′)	Annealing Temperature	Product Length	Amplification Efficiency
*β-actin*	F: CTCCCTGGAGAAGAGCTACG R: TTGAAGGTGGTCTCGTGGAT	59.18 °C 60.51 °C	138 bp	104%
*GAPDH*	F: TCGGAGTGAACGGATTTGGC R: GCCGTGGGTGGAATCATACT	60.67 °C 59.82 °C	146 bp	105%
*BCO1*	F: ACGCGACCTCAGAGACAAAT R: TGAAAACGTTTCCAGCAGCG	59.40 °C 59.97 °C	141 bp	102%
*BCO2*	F: GGCTGTGGTTTTCGGCATTT R: GCTCCTGGTACTGGCACAAA	59.97 °C 60.25 °C	128 bp	89%
*LRAT*	F: ATGGGCCTGGCATCCTATAC R: CACAGTTGACGTGGGGAAAG	59.00 °C 59.06 °C	93 bp	115%
*TTPA*	F: CCCAGACATTCTTCCTCTGG R: ATGAATGGGCTCAGAAATGC	59.65 °C 60.04 °C	124 bp	113%

## Data Availability

The authors declare that the data of this research are not deposited in an official repository and data will be available upon reasonable request.

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
