# Peer review of "The Effect of the BCO2 Genotype on the Expression of Genes Related to Carotenoid, Retinol, and α-Tocopherol Metabolism in Rabbits Fed a Diet with Aztec Marigold Flower Extract"

_ijms, 2022, doi:10.3390/ijms231810552_

Round 1

Reviewer 1 Report

The authors deal with a topic of great interest in an analytically and exhaustive way.

The experiments are described with care and relevance, the supporting literature cited is satisfactory and the conclusions are centered. For my part, I express total satisfaction and consider the work worthy of being published on IJMS without any other changes

Author Response

Dear Reviewer,

Thank you for perusal of our manuscript. We are very glad that you liked its content. We hope the findings will contribute to a better understanding of carotenoid metabolism in rabbits and possibly other mammals.

With best regards

authors

Reviewer 2 Report

In the presented manuscript, Strychalski J. et al. have analyzed the correlation between BCO2 polymorphism and expression of BCO1, BCO2, LRAT, and TTPA genes in liver samples of rabbits. 

The introduction is well written, but this section needs to be shortened to the most critical information introducing the reader to the topic. 

It would be nice to confirm the obtained results on the cell lines. 

The discussion section is hard to read and needs to be improved. Please try to rearrange this section for better understanding and make it easier to follow the researcher's thoughts. The discussion is written inconsistently, which makes it difficult to read. It should also indicate which information in the literature is consistent and contrary to the obtained results. Also, the discussion section lacks a part with the limitation of this study. In the discussion section, literature concerning the analyzed factors is nicely cited.

Details:

Table 1 - the table is shown on page three, while the description is on page ten. The Authors should be more diligent when preparing the MS.

Table 2 - should be placed in the methods section.

Figure 1 and Figure 2 - must be placed earlier in the article. It is hard to read in the presented form.

Tables 3 and 4 - reconsider how to write "p - value" - p should be italic. Check the whole article in this regard.

I am missing the information about the potential use of this research - please provide how those findings may be implicated in the future.

The conclusion sections need to be improved. The Authors can not duplicate the sentences. 

Author Response

Dear Reviewer,

Thank you for thorough perusal of our manuscript. All comments were addressed during the revision process, and the manuscript was revised accordingly. As per editorial requirements, all changes were marked in the revised manuscript using the “Track Changes” mode. Below are our responses to the Reviewer’s comments.

Reviewer’s comment:

The introduction is well written, but this section needs to be shortened to the most critical information introducing the reader to the topic.

Response:

- Thank you for this suggestion. The Introduction section was shortened.

Reviewer’s comment:

It would be nice to confirm the obtained results on the cell lines.

Response:

- We agree with the Reviewer. Unfortunately, cell lines were not used in our study. We have always relied on farmed rabbits in the production cycle, which has many advantages, including a direct reference to practice. However, in view of the Reviewer’s opinion, a suggestion for future research involving cell lines was included in the revised manuscript, in the last paragraph of the Discussion section.

Reviewer’s comment:

The discussion section is hard to read and needs to be improved. Please try to rearrange this section for better understanding and make it easier to follow the researcher's thoughts. The discussion is written inconsistently, which makes it difficult to read. It should also indicate which information in the literature is consistent and contrary to the obtained results. Also, the discussion section lacks a part with the limitation of this study. In the discussion section, literature concerning the analyzed factors is nicely cited.

Response:

- The Discussion section was rearranged so that the results are discussed following the sequence presented in the M&M section. Our findings were also compared with the results of previous research. Finally, the last paragraph was added to the Discussion section to describe the limitations of the study and the directions for future research. Thank you for the insightful comments that helped us significantly improve the manuscript.

Reviewer’s comment:

Table 1 - the table is shown on page three, while the description is on page ten. The Authors should be more diligent when preparing the MS.

Response:

- We apologize for this oversight. The tables were moved and numbered accordingly.

Reviewer’s comment:

Table 2 - should be placed in the methods section.

Response:

- Table 2 was placed in the Methods section, as suggested by the Reviewer.

Reviewer’s comment:

Figure 1 and Figure 2 - must be placed earlier in the article. It is hard to read in the presented form.

Response:

- The relevant changes were made.

Reviewer’s comment:

Tables 3 and 4 - reconsider how to write "p - value" - p should be italic. Check the whole article in this regard.

Response:

- Thank you for this remark. The "p" was italicized in Tables 1-4 and throughout the manuscript.

Reviewer’s comment:

I am missing the information about the potential use of this research - please provide how those findings may be implicated in the future.

Response:

- As suggested by the Reviewer, the implications and limitations of the study, as well as the directions for future research were added at the final paragraph of Discussion.

Reviewer’s comment:

The conclusion sections need to be improved. The Authors can not duplicate the sentences.

Response:

- Thank you for this suggestion. In the revised manuscript, the conclusions are presented in the last paragraph of the Discussion. We also improved the conclusions in the Abstract.

 Because you have suggested that the manuscript should undergo extensive English revisions, the revised manuscript has been edited by a professional translator (Aleksandra Poprawska) who has extensive experience in editing scientific manuscripts. The manuscript has also been spell-checked and grammar-checked by a native English speaker.

Once again, we would like to thank the Reviewer for all valuable comments which have led to significant improvements in the manuscript. We hope we responded to each comment with due care.

Round 2

Reviewer 2 Report

Accept in present form.